# Experimental Release of Orphaned Wild Felids into a Tropical Rainforest in Southwestern Costa Rica

**DOI:** 10.3390/vetsci9090468

**Published:** 2022-08-31

**Authors:** Víctor H. Montalvo, Isabel Hagnauer, Juan C. Cruz-Díaz, Brayan Morera, Kevin Lloyd, Carolina Sáenz-Bolaños, Todd K. Fuller, Eduardo Carrillo

**Affiliations:** 1Instituto Internacional en Conservación y Manejo de Vida Silvestre, Universidad Nacional, Apdo, Heredia 1350-3000, Costa Rica; 2Namá Conservation, Heredia 40101, Costa Rica; 3Rescate Wildlife Rescue Center, Fundación Restauración de la Naturaleza. Apdo, 1327-4050 Alajuela, Costa Rica; 4Amazon Conservation Team, Arlington, VA 22203, USA; 5Department of Environmental Conservation, University of Massachusetts, Amherst, MA 01003, USA

**Keywords:** *Leopardus*, reintroduction, rescue, felid, telemetry, post-release monitoring, ocelot, margay, tropical rain forest

## Abstract

**Simple Summary:**

A male ocelot (*Leopardus pardalis*) and a female margay (*Leopardus weidii*) brought in from the wild were held in captivity and rehabilitated, then radio-collared, released, and monitored at a national wildlife refuge previously assessed for predator and prey occurrence. Subsequently, the ocelot was trapped while preying on chickens, and the margay was found dead, likely due to ocelot predation. Avoiding habituation to humans, ensuring hunting abilities, and assessing release sites likely is not sufficient to ensure successful release of these species.

**Abstract:**

A 3- to 4-mo-old male ocelot (*Leopardus pardalis*) and a 6- to 8-mo-old female margay (*Leopardus weidii*) were brought in from the wild, held in captivity, and rehabilitated for 906 and 709 days, respectively, at the Rescate Wildlife Rescue Center in Costa Rica. During captivity, both cats were kept as isolated as possible from humans and fed appropriate live wild prey. After maturing and demonstrating the ability to capture and feed on live prey, the cats were radio-collared, released at a national wildlife refuge previously assessed for predator and prey occurrence, and monitored. After 54 days, the ocelot was trapped while preying on chickens in a nearby community, and after 20 days, the margay was found dead, likely due to ocelot predation. Avoiding habituation to humans, assuring hunting abilities, and assessing release sites likely is not sufficient to assure successful release of these species, and more experimental releases with innovative and detailed protocols and monitoring are needed.

## 1. Introduction

Carnivore populations around the world are declining due to significant anthropogenic impacts on most of Earth’s ecosystems [1,2]. Since carnivore species are keystone organisms [1], translocations, reintroductions, and other releases have been implemented at local and global scales to ameliorate their diminution [3,4]. These actions often involve an attempt to release individuals born in captivity or rehabilitated from natural populations to re-establish or supplement a species population within their natural range [5,6,7].

Veterinary treatment and rehabilitation of indigenous wildlife brought into captivity often leads to animals, perhaps millions each year, being released back into the wild [8]. In areas with unique indigenous species, the care and release of individual animals would seem an important part of conservation [9,10], but, in fact, few studies of such releases have demonstrated population benefits [11,12]. Still, given the efforts made and the number of animals released, evidence-based, species-specific, and context-specific protocols need to be developed to ensure that wildlife survival is maximized during rehabilitation and post-release [8]. This is especially true because natural disasters and extreme weather events are escalating globally, in part due to climate change [8]. In addition, public education is often viewed as an important aspect of release programs, as it may contribute indirectly to conservation by increasing awareness [11].

Most attempts to re-establish self-sustaining populations of carnivores are experimental and therefore need a holistic approach focusing on biological, technical, valuational, and organizational aspects [13]. At a minimum, released animals should be able to find prey, reproduce, and avoid predators and potentially dangerous species, such as humans [7,13,14,15]. Releases of felids into the wild, in particular, are difficult, and there is limited information on whether they fail or succeed [16,17,18,19]. Records providing baseline data against which the success/failure of the releases can be measured are crucial to conservation practices.

Here, we document the release efforts for an ocelot (*Leopardus pardalis*) and a margay (*Leopardus weidii*), both orphans born in the wild, into the tropical forest of Costa Rica. We present their case histories, including our management strategy (in/ex situ), a summary of dietary items (ex situ), and post-release movement patterns and fates (in situ).

## 2. Materials and Methods

### 2.1. Rehabilitation and Release Location Sites and Management

The Rescate Wildlife Rescue Center in Alajuela, under the authorization of the Ministry of Environment of Costa Rica, rehabilitates orphaned, injured, or confiscated wild animals and either releases them back into the wild or, if they would not survive in the wild for health or behavioral reasons, keeps them in the sanctuary for public display and education. Most species brought to rescue centers come from authority confiscations (46%) and unnecessary rescues (20%). Once brought to the center, a complete veterinary assessment is performed to confirm their age and health status, and a preliminary determination is made regarding the possibility of releasing them back into the wild.

For the case studies noted below, individuals were isolated in a 0.4-ha enclosure (Figure 1) with minimal human contact. During the first days of captivity, the cats were observed to see if they acted wild (aggressive, avoiding humans, and in search of shelter), and the decision was made to eventually release them, but first to feed them small living prey items, thus providing a behavioral necessity for potential release. To do so, living prey was released into the pentagon shape enclosure from a different feeding gate (5 feeding gates in total) at once and at random schedules, leaving 3–5 day intervals with no food to simulate wildlife conditions. Additionally, to prevent the association between humans and food arrival, the enclosure perimeter was covered with agricultural shade netting (Figure 1).

The release site, the Centro de Investigación para la Conservación de Fauna Silvestre San Josecito (CISJ), is located in the Golfito district of Puntarenas Province, southwestern Costa Rica (8°38′44.24″ N, −83°50′39.26″ W), and is comprised of a mosaic of human activities such as livestock farms, oil palm and wood plantations, and small self-subsistence farms; it also shares a boundary with Piedras Blancas National Park (PBNP). The weather is hot and humid, with an annual rainfall of 3000 mm, peaking from June to September, and an average temperature of 25 °C [20].

### 2.2. Pre-Release Preparation

Prior to the releases, we deployed five camera traps in the CISJ near the release site as an exploratory effort to assess the abundance of potential wildlife species that might influence the sustainability of medium–small-size felid populations (i.e., prey and predator abundances). To standardize photo rates and counts in independent photos per 100 trap nights, we used Montalvo et al.’s [21] methodology to define photo-capture rates.

### 2.3. Release Protocol

A week before being released, each cat was chemically immobilized using a dart projectile (DANiNJECT©; Kolding, Denmark) with a combination of 5 mg/kg of ketamine (10% ketamine; Bremer Pharma GmbH) mixed with 0.5 mg/kg xylazine (Procin Equus 10%, Pisa Agropecuaria) [22], and fitted with a satellite telemetry unit (Lotek©; Lite Track Iridium150, Newmarket, ON, Canada; Telonics©, TGW-4170-4, Mesa, AZ, USA), following the handling procedure guidelines of the American Society of Mammologist [23], with the approved permission of the Environmental Minister of Costa Rica (SINAC-ACOSA-DT-PI-INV-030-2020). Each telemetry collar was customized and weighed an average of <3 % of the individual’s total mass. Four days before a hard release, each individual was transported 500 km by vehicle from the rehabilitation center to the release site in a wooden box (1.2 m × 0.5 m × 0.5 m), and then put into a 0.2-ha pre-release enclosure for two days in order to acclimatize them to the area and make a final assessment of the individual’s fitness for release into the wild.

### 2.4. Post-Release Monitoring

To evaluate post-release movements, we used location data to estimate and plot movement patterns using the statistical software R version 4.1.3 [24] with the package “adehabitatHR” [25]. Specifically, for space use estimation, we used the 100% Minimum Convex Hull (MCH) method and calculated the step dispersal distance after plotting and connecting the sequential locations for each individual.

## 3. Results

### 3.1. Preliminary Wildlife Assessment

With a total effort of 1298 trap nights, 16 species were identified (14 mammals, one bird, and one reptile). The highest photo-capture rates (Figure 2) were for agouti (*Dasyprocta punctata*; 53.4 captures/100 TN), collared peccary (*Pecari tajacu*; 40.7), and great curassow (*Crax rubra*; 16.2), while the lowest rates were registered for jaguarundi (*Puma yagouaroundi*; >0.1), black iguana (*Ctenosaura similis*; >0.1) and paca (*Cuniculus paca*; >0.2). Ocelots, but no margays, were also photographed.

### 3.2. Case 1; Ocelot ♂

In October 2019, during the wet season, a 3- to 4-month-old ocelot (2.36 kg, ♂), was rescued after domestic dogs chased the mother away; it was kept by a family for 2 days in the Portalón district of Puntarenas Province, southwestern Costa Rica (9°22′3.3″ N, −83°57′50.8″ W), then confiscated by park rangers and moved to Rescate Wildlife Rescue Center facilities. We determined after 6 weeks of observation and care in isolation that this individual was a candidate for release due to the wild behavior it exhibited (e.g., aversion to humans, ability to capture and subsist on wild prey). This ocelot remained in captivity for 23 months, by which time it was able to chase and kill any prey released into the enclosure, consuming 76% of mammal biomass (100 g to 5 kg; Table 1, Appendix A), 4.5% reptile biomass (1 kg to 5 kg), and 19.5% bird biomass (100 g to 5 kg). A week before being released, this male ocelot weighed 12.1 kg and was fitted with a satellite telemetry collar programed to record daily locations every 8 h; additionally, a final veterinary assessment was performed. This individual was released in January 2021 at the site of CISJ.

We collected a total of 93 locations (57% of potential attempts) during 54 post-release monitoring days (15 January–10 March) with a mean of ~1.7 locations per day (Table 2). During the 54 days, this ocelot ranged over 44.4 km^2^ (Table 2, Figure 3), with an average step dispersal distance of 0.48 km (max distance moved in straight line from the release site; 11 km). On day 54, this ocelot was live-trapped by locals 10 km from the release site in a nearby community outside the wildlife refuge because it was preying on backyard chickens. The animal was recovered by the firefighting department of the Golfito district and was taken back to the Rescate Wildlife Rescue Center facilities for veterinary evaluation. The cat had lost 3.4 kg (weight 8.7 kg, which is still in the mean range reported for Costa Rica: 8–10 kg) and had minor injuries (rubs and small wounds probably due trapping) that had occurred since the release, but otherwise it was deemed healthy and subsequently re-released in the CISJ without the satellite telemetry unit.

### 3.3. Case 2; Margay ♀

In December 2020, during the dry season, a 6- to 8-month-old margay (1.3 kg, *♀*), was found in the Vara Blanca district of Heredia Province, Costa Rica (10°10′8.4″ N, 84°07′51.0″ W). After assessing the potential for release, this margay stayed in captivity for 14 months and became able to chase and kill any prey released into the enclosure, consuming 61.1% mammal biomass (<100 g; Table 1, Appendix A), 0.9% reptile biomass (200 g to 500 g), and 38% bird biomass (200 g to 500 g). A week before being released, this margay weighed 2.9 kg and was fitted with a satellite telemetry unit programed to record daily locations every 8 h. After a final veterinary assessment, this individual was released in January 2022 at the site in Golfito. We obtained a total of 13 locations (22% of potential) during 20 post-release monitoring days (20 January–10 February) with a mean of ~0.6 locations per day (Table 2). During the 20 days, this margay ranged over 3.48 km^2^ (Table 2, Figure 3), with an average step dispersal distance of 1.17 km (max distance moved in a straight line from the release site −5 km). On Day 20, the mortality alert in the collar was active, and after corroborating the last cluster of locations on the field, we found the margay carcass 5 km away from the release site with four punctures in the skull, subsequently suggesting bite marks (length 66 mm, width 42 mm) caused by canine teeth of an ocelot.

## 4. Discussion

Releasing wild felids into the wild is a challenging conservation effort [16,17]. Animals need to re-integrate into the wild for a release to be considered truly successful [26], and this includes normal behavior and future breeding. Behavioral development before release (e.g., regarding hunting skills) is particularly important for helping to ensure that animals survive [27], as are other but more difficult skills to acquire, such as predator evasion, interaction with conspecifics, finding shelter, moving in complex terrain, and demonstrating orientation and navigation in complex environments [8,10,12,28].

There are a variety of ways to define the success of a release, including having an established home range, one-year survival, and/or breeding in the wild [8,10,12]. That these are difficult markers to achieve was demonstrated by an early (1992) margay and ocelot release project in Costa Rica prompted, in part, by a hurricane that destroyed a rescue center and resulted in the unplanned release of a number of both species of cat [28]. Subsequently, previous habituation, weight loss, depredation events, mortality in the wild, and interactions with predators all hindered the success of releases.

Our results indicated that the ocelot survived 54 days post-release and used a range that was larger than in previous studies of wild ocelots in Central America (~18.9 km^2^) [29], suggesting it was still in search of a permanent place to settle; it was then recaptured in a backyard chicken coop (cf. Hayward et al. [16] as reported for large felids in South Africa). With regard to the weight loss reported for this ocelot individual, it was still on the mean reported for Costa Rica (8–10 kg). Hence, we speculate that long-range movements and high activity would cause this weight loss.

We do not know the fate of this ocelot after being released a second time, and though it did show the ability to kill wild prey and survive, we believe that additional site aspects, such as the surrounding landscape and the abundance of competitors, played an important role in affecting release success [7,30,31,32,33]. Since ocelots were the felid species with the highest photo-capture rates in our release area, we speculate that resident individuals could marginalize and exclude new individuals from occupied territories, thus forcing reintroduced individuals to explore new areas outside the refuge. Despite the fact that the margay exhibited the ability to kill wild prey, it apparently lacked the basic skills to avoid potential predators, leading to its death by an ocelot [30,31,32,34,35]. Specifically, ocelot predation on margays is rare [27]; nevertheless, poor body condition combined with a lack of predator avoidance skills could lead to such an event for a reintroduced animal.

The interpretation of our results should be used cautiously to capitalize on the most important insights. We recognized as critical the reduction of human contact to avoid human imprinting behavior during the ex situ training, as well as a proper wild diet with prey species potentially found in the release area. In both of our cases though, the evidence suggests the capability to navigate and locomote within new environments is crucial [13], and perhaps more acclimatation (e.g., a longer, soft-release with an on-site pen facility) might work better than a hard release so that rehabilitated individuals could better recover from the handling and transport stress, and to allow individuals to adjust to local conditions [34,35,36]. Additionally, during the ex situ training of both cats, they each obtained the skills to kill wild prey, but we did not provide any training that would help them avoid predation—we identified as challenging the teaching of “danger-avoidance behaviors” with predators, conspecifics, and humans to evade lethal encounters in new environments. As it turned out, the individual responses in the field were different—one individual was killed by a predator, and the other had lost weight and was caught preying on backyard chickens outside the protected area.

Though a few other such studies have included preliminary assessments of prey [4], habitat, movements, and local human population [19], we recommend an adequate evaluation of biological, social, and environmental factors for each release location, followed by a post-release monitoring program to help understand the movement, dispersal, and mortality causes of reintroduced felids. Despite the several protocols and guidelines available for animal releases and reintroductions [5,8,11], there is not a universal methodology that always works because releases deal with large amounts of uncertainty [35,37,38]. Although our information is limited to only two individuals, and it is likely that they did not represent a major contribution in terms of population dynamics, the experience we gained and the lessons we learned and share are valuable for small cat conservation management using a technique for which field evidence is limited. For small felids, more experimental evidence is needed, and post-release monitoring is critical in places where reintroducing rehabilitated animals is common practice. We cannot ignore an animal’s post-release fate, and whether such releases are a success or failure, it is important to learn what went right and wrong so that pitfalls and shortcomings in future carnivore releases can be avoided.

## Figures and Tables

**Figure 1 vetsci-09-00468-f001:**
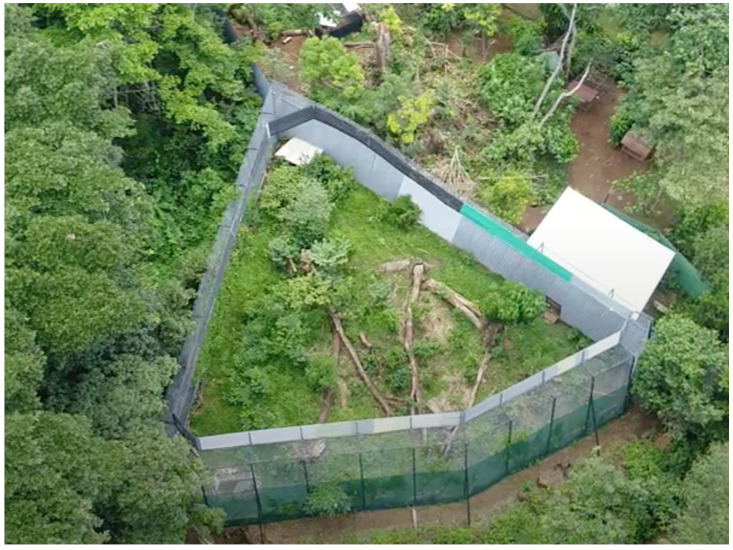
Pre-introduction enclosure in the Rescate Wildlife Rescue Center facilities: 0.4-ha area with a pentagonal shape and a 20% slope, 6-m high fence totally covered with 6 × 6 cm mesh galvanized wire, and double covered in the outside with 2-m high agricultural shade netting. To help acclimatize the cats, the interior had a 1.5-m × 1-m pond, logs as shelters, and 6 tree species of 4.5-m maximum height.

**Figure 2 vetsci-09-00468-f002:**
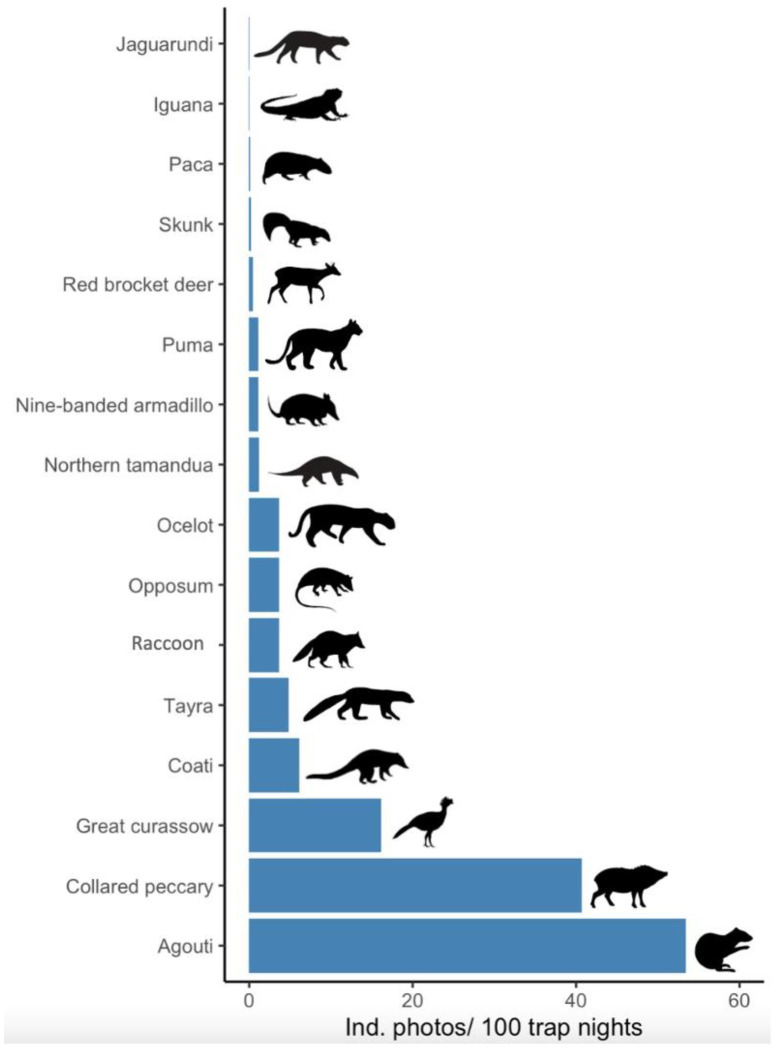
Photo-capture rates (no. of independent photos per 100 trap nights) of common large vertebrate species registered near the animal release site in the Osa Conservation Area, Golfito, Costa Rica.

**Figure 3 vetsci-09-00468-f003:**
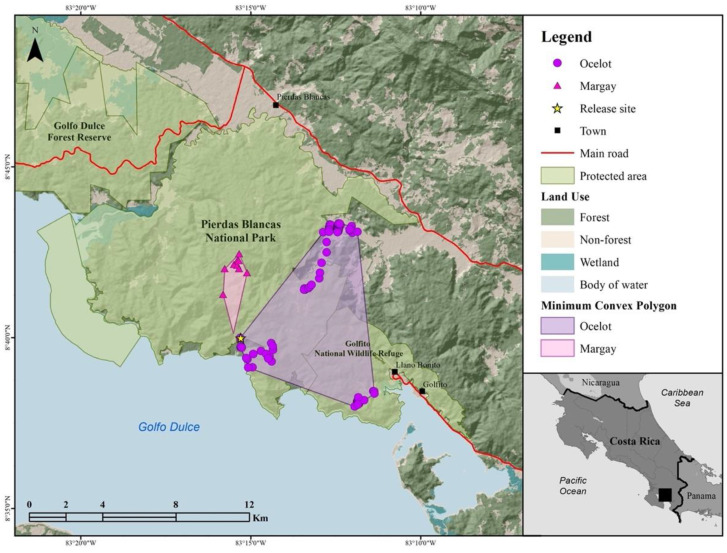
Locations and 100% minimum polygon delineation of a male ocelot and female margay after release from captivity in the Osa Conservation Area, Golfito, Costa Rica.

**Table 1 vetsci-09-00468-t001:** Total amount of live prey biomass in weight classes offered to a male ocelot (*Leopardus pardalis*) and a female margay (*Leopardus weidii*) during 906 and 709 days of enclosure in the Rescate Wildlife Rescue Center before their release into the wild in Alajuela, Costa Rica.

	Ocelot		Margay	
	Percent of Total	Prey Items (*n*)	Percent of Total	Prey Items (*n*)
Reptile biomass				
200–500 g	—	0	0.9	6
1–5 kg	4.5	41	—	0
Bird biomass				
<100 g	—	0	5.2	37
100–200 g	5.5	50	10.4	74
200–500 g	12.7	115	18.5	131
500–1000 g	0.3	3	3.9	28
1–5 kg	0.9	8	—	0
Mammal biomass				
<100 g	25.6	232	46.3	328
100–200 g	29.0	263	13.0	92
200–500 g	16.4	149	1.8	13
500–1000 g	4.1	37	—	0
1–5 kg	0.9	8	—	0

**Table 2 vetsci-09-00468-t002:** Post-release movement and space use parameters of a male ocelot and a female margay after release from captivity into the Osa Conservation Area, Golfito, Costa Rica.

	Ocelot	Margay
Release age (months)	19	22
100% Minimum Convex Hull (MCH) (km^2^)	44.38	3.48
Step dispersal distance (km)	0.484; CI [0.212–0.757]	1.171; CI [0.379–1.963]
Tracking days (n)	54	20
Locations (n)	93	13

## Data Availability

All data presented in this study are available in the text and tables.

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
