# Peer review of "Experimental Release of Orphaned Wild Felids into a Tropical Rainforest in Southwestern Costa Rica"

_vetsci, 2022, doi:10.3390/vetsci9090468_

Round 1
Reviewer 1 Report
You've cast this in terms of the reintroduction of felids, but it is unclear how the release of two captive-raised felids in isolation meets the IUCN definition for a reintroduction, i.e. an attempt to re-establish a viable population within a part of the indigenous range of the species from which it has been extirpated. It is unclear if the release areas were within the indigenous range, or if conspecifics were still present - this last point is critical since both animals were captured when young (before independence?) and might not have been well socialised nor have all requsite survival skills. Releasing single rehabiliated animals too does not seem to be likely to make a major contribution to species conservation. One animal died and the other was losing weight when last assessed - the bottom line seems to be that the rehabilitation release (best to call it that rather than a reintroduction) failed in both cases. Really all you can then conclude is that despite the animals learning to capture live prey in a captive enclosure, they were unable to sustain themselves when free ranging. Possibly even the apparent predation of the margay might have been facilitated by poor condition. Good to assess the prey base in the release area, but this needs context - what is normal or sufficient for these species? Good also to attempt post-release monitoring, but this should be tightly focussed on addressing a priori questions rather than broadly to track movements. The addition of GPS tags only a week before release might have imposed a significant burden on both animals. Good to document these releases somewhere - but best not to make too much of it in terms of wider learnings for the true reintroduction of felids.
Author Response
See the attached document

Reviewer 2 Report
Montalvo et al. have rehabilitated 2 orphaned felids and monitored their post-release survival. While this work is opportunistic and anecdotal it is nonetheless an important account that highlights the perils of reintroduction biology. I think this is important information that should be published, I also think there are opportunities for improving the clarity and presentation.
I suggest a much changed introduction. The introduction explores reintroduction biology, but these accounts are actually rehabilitations of wild-born individuals. The role that rehabilitated individuals play in translocation biology is an important and controversial topic. The authors should explore this topic in the introduction to set up this article. The work here is really a documentation of population augmentation with rehabilitated individuals rather than reintroduction.
L34: re-establishment
L51: I suggest starting the Materials and Methods section with a section describing how the individuals used in the study came to you. These are currently presented separately much later in the manuscript and is not an effective way to set up the study.
Section 2.1: Opportunity here to discuss confiscation. Is this a common occurrence? Please provide context.
L67: minimal
L70: What prey species were provided? How was it decided what would be fed at any given time?
L71: shaped
L77: double
L78: I don't think "conditioned" is the correct term. also "pound"? pond?
L80: need to be more clear about what you are assessing with the camera traps. You are exploring the prey species that occur and their relative abundances (or detection rates). You cannot claim to be "determining whether the abundance of wildlife was suitable to support" the felids, unless you are comparing your relative abundances vs a baseline that is known to be needed.
L84: released
L80-90 - this should be 2 paragraphs. Why would you combine the anesthesia methods with the camera trap methods in one paragraph.
L91-92: Insert a section for release protocol. I take it this was hard release. How far away from the rehab pen was the release site. Why was it chosen and how was the animal transported there? Was it released immediately after anesthesia and collaring? These are important points for reader to understand and especially because you revisit some of these decisions in the discussion
L96: please also provide the total straight line dispersal distance (or range length). This is an important space use variable for translocated wildlife because it provides an idea how far away from release site they are going and the likelihood they will move outside of reserves they are released into.
Figure 2: Very cool figure. Spelling of Jaguarundi (figure) vs Yaguarundi (legend)
L109: strongly suggest moving the introduction of the "case studies" to the start of the materials and methods. Its jarring to not receive this information until the Results section.
L114-115: based on what? What does this mean?
L115: How was the time kept in captivity determined?
L129: straight line dispersal distance would be a valuable addition
L134: Was the weight loss within a normal range or was this concerning?
L134: provide further detail about injuries? From the trapping?
Figure 3: another very nice figure
L171-172: Not sure what you mean here. Please re-word for clarity. Not sure you can make any inference about competitors. I think its fair to point out that its not uncommon for reintroduced wildlife that are unfamiliar with their surroundings to prey on livestock or human-subsidized foods like you observed here, I don't think that has anything to do with competitors but rather unfamiliar animals at a disadvantage and having to go for the easy (but ultimately dangerous) prey.
L174: is it unusual for margay to be depredated by Ocelot? Do Margary have high annual survival rates? Some context would be helpful.
L178: good points but this emphasizes that no information was provided to reader about release method / details
L182: although the first animal had been losing weight and preying on unnatural prey.
Author Response
See the attached document

Reviewer 3 Report
I appreciate this case study and think it would be a good addition to the literature. There are however some areas that need to be improved prior to publication.
Namely in the, methods sections and the individual case studies. There is no mention of the actual type of prey and the correlation to those they would find once released. You need to describe the prey items you offered and how the cats were able to capture them. There is biological relevance to “how” one catches what types of prey that can help or hinder success in the wild. Were they simply cornering the terrestrial mammalian prey in the corner of the enclosure, or did they employ stalk, ambush, kill strategies?
We need more descriptive information about their overall behavioral repertoire while in captivity, to glean any lessons from this paper. Considering the Ocelot had significant weight loss, and was found around domestic livestock (twice?) I could argue that it was unable to hunt.
Where I see this paper as being helpful is in presenting the two cases more completely, with a clear description of how you could use that knowledge to improve the hunting and anti-predator (respectively) conditioning prior to release.
Author Response
See the attached document

Round 2
Reviewer 1 Report
Thank you for your revision of your ms. I do still have some issues with it.
First, you have included the framing statement: “Other researchers and agencies conceptualized reintroductions with a broad scope, therefore including the returning of a species into several locations where the species is capable of survive and not only where they have been extirpated”
This is frankly just wrong – if conspecifics remain at the release site, then any release is not a reintroduction, by definition. Potentially a given reintroduction project might release animals in several phases, so conspecifics from earlier releases might be present during subsequent releases. This does not apply to your situation.
Also important is that for any conservation translocation, conservation benefit is critical. The conservation benefit might relate to species status and/or ecosystem functions. It is hard to see how the release of a single young animal could be a contribution to the conservation status of the species.
I strongly suggest reframing as an experimental rehabilitation release, with a view to perhaps learning more to inform future releases of these species, whether these are for future reintroductions or individual animal rehabilitation releases. Trying to cast this as a conservation translocation, let alone a reintroduction, will just highlight a series of problems of definition and scope.
Author Response
Responses to Comments:
We now understand and completely agree with the reviewers’ suggested approach and wording regarding release of rehabilitated animals back into the wild. We have retitled the manuscript, rewritten the introduction, changed the discussion, and added new references. More specifically, we have avoided using the word “reintroduction” with regard to these 2 cases and other such releases.
To address the problems of the presentation of information in the Study area and Methods sections, we have re-written the text to first present generic information relative to both case studies, then follow with the case studies (i.e., from the general to the specific). This should make the manuscript more readable.
Regarding the specific prey items, we now include a list of the variety of species used as an appendix.
Reviewer 2 Report
The authors have improved the manuscript although I still see room for additional improvement. Some of my larger suggestions for improving presentation were not included. Primarily, I still think the manuscript would be improved by shifting the focus from broadscale wildlife reintroduction to discussing rehabilitation of individuals as a way to augment populations. At least one of the other reviewers felt this way too. The authors argue that reintroduction encompasses rehabilitation, which is true, but a more clear focus on rehabilitation is a more clear and honest way in which to frame these 2 case studies.
At least 2 of the reviewers thought it would be valuable to understand the prey items offered to animals but this hasn't been included. Its unclear why this has to be protected and withheld.
I still feel like the flow of the manuscript is compromised by going through the Introduction, Methods, starting results and then having a section for the case studies which represents a narrative step backwards. Text such as "In October 2019, during the wet season, a 3- to 4-month-old ocelot (2.36 kg, ♂), was 331 rescued after domestic dogs chased the mother away; it was kept by a family for 2 days in 332 the Portalón district of Puntarenas Province, southwestern Costa Rica (9°22'3.3"N, - 333 83°57'50.8"W), then confiscated by park rangers and moved to Rescate Wildlife Rescue 334 Center facilities. We determined after 6 weeks of observation and care in isolation that this 335 individual was a candidate for reintroduction" seems to me like it should be provided much earlier (end of introduction or early methods) as opposed to the middle of the results. This is less important though.
Author Response

(The authors gave the same response as above.)
